# Effects of Watermelon Cropping Management on Soil Bacteria and Fungi Biodiversity

Mei Tian [1,†], Jinjin Liang [2,†], Shengfeng Liu [1], Rong Yu [1] and Xingxu Zhang [2,*]

1   Institute of Horticulture, Ningxia Academy of Agricultural and Forestry Sciences, Yinchuan 750002, China
2   College of Pastoral Agriculture Science and Technology, Lanzhou University, Lanzhou 730020, China
*   Correspondence: xxzhang@lzu.edu.cn; Tel.: +86-138-9360-0874
†   These authors contributed equally to this work.

**Abstract:** Watermelons grown in sandy soil are rich in trace elements, particularly selenium, and are therefore also known as selenium-rich sand watermelons. However, continuous watermelon cultivation in the same sandy field decreases soil fertility and degrades the ecosystem, ultimately resulting in low-quality watermelons. Introducing different crops into the crop pattern could alleviate the problems posed by continuous cropping. A field experiment was conducted to explore the effects of different crop patterns on soil microbial communities and soil properties via standard techniques. The results showed that 14,905 bacterial and 2150 fungal operational taxonomic units were obtained and assigned to eight bacterial and five fungal phyla, respectively. Soil bacterial communities primarily comprised Proteobacteria, Planctomycetes, Actinobacteria, and Acidobacteria, and the soil fungal community was dominated by Ascomycota, Chytridiomycota, and Basidiomycota. Different crop patterns had a significant effect on the Chao and ACE indexes of fungal communities in the soil. The rotation of six years of watermelon and one year of wheat had the highest richness indexes of all the rotations. Different crop patterns had significant effects on soil properties, such as organic matter (OM), total nitrogen (TN), total potassium (TK), available phosphorus (AP), available K, nitrate nitrogen (NN), and pH. The soil OM, TN, NN, and pH of six years of watermelon and one year of wheat cultivation were significantly higher than those of the other three crop patterns. In addition, the soil TK and AP of the continuous watermelon planting treatment were significantly higher than those of the other three crop patterns. Redundancy analysis results revealed many complex relationships between soil properties and soil bacterial or fungal communities. Employing different crop patterns plays an important role in the effective regulation of soil microbial diversity and properties.

**Keywords:** agricultural; crop management; diversity; microbial community; watermelon

## 1. Introduction

Watermelon (*Citrullus lanatus*, 2n = 2x = 22), native to Africa, belongs to the genus *Citrullus* of the family Cucurbitaceae and is one of the most widely grown fruit crops globally [1]. The latest classification of the genus *Citrullus* contains seven extant species: *C. amarus*, *C. colocynthis*, *C. ecirrhosus*, *C. lanatus*, *C. mucosospermus*, *C. naudinianus*, and *C. rehmii* [2,3]. According to the FAO, 3,053,258 ha of watermelon were planted worldwide in 2020, producing 101,620,420 t (https://www.fao.org/faostat/en/# accessed on 1 January 2020). In China, the planting area and production in 2020 were among the highest globally at 1,405,871 hm$^2$ and 60,246,888 t, respectively. Watermelon is particularly popular during summer because it is rich in water and nutrients, including vitamins A, B, and C and lycopene [4].

Farmers growing plants in a mono-cropping system are troubled by continuous cropping, leading to yield reduction and quality decline [5]. The yield of worldwide main food crops, including wheat (*Triticum aestivum*), rice (*Oryza sativa*), corn (*Zea mays*),

and soybean (*Glycine max*), decreases under continuous cropping [6–8]. Yield losses in continuous cropping can be attributed to plant and microbial interactions in the soil [9]. For example, dynamic changes in bacterial and fungal populations may be important factors in reducing peanut growth and yield over many years of continuous cropping [10,11]. Continuous field pea (*Pisum sativum* L.) cropping has a negative impact on crop yield, soil organic matter (OM) levels, and soil microbial community structure and function [12]. However, breaking continuous cropping through different strategies, such as interplants, rotations, and plant growth-promoting rhizobacteria, could alleviate its shortcomings. For example, a watermelon monocrop intercropped with aerobic rice alleviated *Fusarium* wilt in watermelon by restraining *Fusarium* spore production and altering the microbial communities in the rhizosphere soil [13]. The potato-legume rotation system can improve the soil bioenvironment, alleviate continuous cropping obstacles, and increase potato tuber yield in semiarid regions [14]. Moreover, plant growth-promoting rhizobacteria alleviate aluminum toxicity and bacterial wilt in ginger (*Zingiber officinale* Roscoe) in acidic continuous cropping soil [15].

The gravel-sand mulched field, also known as the "sandy field" in Chinese, is a traditional tillage pattern in the semiarid Loess Plateau of northwest China, created by farmers in Gansu Province approximately 300 years ago [16,17]. The mulch, a 5–16 cm-thick layer of gravel or pebbles interspersed with coarse sand, is used to conserve sporadic and limited rainfall for reliable crop production [18,19]. Sandy fields in China are primarily distributed in areas with annual precipitation of 200–400 mm, such as Gansu, Ningxia, Xinjiang, and Qinghai [20]. Planting melons and vegetables in sandy fields can improve agricultural production in arid regions and reduce wind and water erosion [21]. Notably, watermelons grown in sandy fields in Zhongwei, China, contain abundant trace elements, particularly selenium, and are also known as selenium-rich sand watermelons. However, continuous selenium-rich sand watermelon cropping is very prevalent in China, causing serious soilborne diseases, decreasing soil fertility and watermelon quality, and harming the sandy field ecosystem [22].

In this study, we set up four crop patterns in a field trial: continuous watermelon cultivation for seven years, a three-year fallow after planting watermelons in the first four years, continuous watermelon–sunflower rotation, and continuous watermelon–wheat rotation. Soil properties and microbial diversity were measured to explore the effects of different crop planting patterns on soil fertility and microorganisms.

We hypothesized that:

(1)　Different crop patterns could affect the chemical properties of soil.
(2)　Different crop patterns could change the soil microbial community composition and diversity.

## 2. Materials and Methods

### 2.1. Site Description and Experimental Design

This study was conducted in field plots in Xiangshan Township, Zhongwei City, Ningxia Hui Autonomous Region, China (105°15′ E, 36°56′ N, altitude 1698 m). The region has a typical north-temperate continental monsoon climate. It is suitable for planting watermelons owing to sufficient light, a highly effective accumulated temperature, and a large temperature difference between day and night. The average annual precipitation is 185.9 mm, and the average annual evaporation is 2000 mm [23]. The annual average sunshine is 2800–3000 h; the annual average temperature is 6.8 °C; the effective accumulated temperature is greater than or equal to 10 °C is 2500–3200 °C; the frost-free period is 140–170 d; and the temperature difference between day and night is 12–16 °C [24].

According to the different watermelon soil rotations, this study included four treatments, and the soil microbial communities were evaluated in four groups Treatment 1 (T1) (planting watermelons for four years from 2013 to 2016, followed by three years of fallow from 2017 to 2019); Treatment 2 (T2) (planting watermelons for seven years from 2013 to 2019); Treatment 3 (T3) (planting watermelons for six years from 2013 to 2018 and then planting wheat in 2019); and Treatment 4 (T4) (crop pattern of one-year watermelon and

one-year sunflower for seven years) (Figure S1). In total, 36 plots (each 5 m × 5 m) were established in 2013.

The experimental field used mulch drip irrigation, and water supplementation once during planting ensured that the field was moist. The models that needed supplemental irrigation were irrigated once at the vine elongation, fruit setting, and expansion stages, 4 L per plant each time, with no supplemental irrigation at other times.

Fertilization was conducted via the hole application method. Briefly, before sowing at approximately 20 cm from the watermelon planting hole, each hole was treated with organic fertilizer (0.4 kg) and bio-fertilizer (~0.1 kg) (effective living bacteria $\geq 0.2$ million/g, OM $\geq 20\%$). There were two fertilizations for each 667 m$^2$ of watermelon: during the watermelon extension period, 20 kg of water-soluble fertilizer (N:P:K = 20:20:20) was applied with water; during the watermelon expansion period, 20 kg of water-soluble fertilizer (N:P:K = 13:5:40) was applied with water.

### 2.2. Sample Collection

Soil samples were collected at the end of plant growth in 2020. For each sub-plot, soil samples were obtained from 20 cm cores, and each of the four treatments had nine soil samples, each mixed from five blended points. The soil samples were cooled, brought back to the laboratory, and stored at $-80\,°C$ in a freezer before DNA extraction. Before chemical analysis, soil samples were screened using a 2.0 mm sieve and stored at $4\,°C$ in a refrigerator, while other samples awaiting analysis were stored at $-80\,°C$ in a freezer.

### 2.3. Physicochemical Soil Analysis

Soil pH was analyzed at a ratio of 1:2.5 in soil/water mixtures. According to the method of Nelson and Sommers (1982) [25], 0.25 mm sieved soil was used to measure soil OM. The ammonium acetate method and flame photometry were used to extract and analyze available potassium (AK) and total potassium (TK), respectively [26]. The molybdenum blue method was used to calculate the plant's available phosphorus (AP) [27]. A continuous flow analyzer (FIAstar 5000Analyzer) was used to measure the total nitrogen (TN), nitrate nitrogen (NN), and total phosphorus (TP) in the soil [28].

### 2.4. DNA Extraction, Amplification, and Sequencing

Total DNA was extracted from approximately 0.1 g and 0.5 g of root and rhizosphere soil samples, respectively, using a plant DNA kit (Tiangen, Beijing) and a soil DNA Kit (OMEGA, Shanghai), respectively, according to the manufacturer's instructions. The V3-V4 region of the bacterial 16S rRNA gene was amplified using the primer pair (335F: 5′-CADACTCCTACGGGAGGC-3′ and 769R: 5′-ATCCTGTTTGMTMCCCVCRC-3′). The fungal internal transcribed spacer 1 (ITS1) region of the rRNA gene was amplified using the primer pair (ITS1F: 5′-CTTGGTCATTTAGAGGAAGTAA-3′ and ITS2: 5′-GCTGCGTTCTTCATCGATGC-3′). Two different thermostable DNA polymerases were used in the 16S rDNA PCR amplifications for each sample to reduce PCR bias: (I) Phusion High-Fidelity DNA Polymerase (Thermo Scientific, Sweden): $98\,°C$ for 2 min followed by 30 cycles of $98\,°C$ for 30 s (denaturation), $56\,°C$ for 20 s (annealing), $72\,°C$ for 20 s (polymerization), and a final extension at $72\,°C$ for 10 min, and the size of the amplified product was confirmed to be appropriate. DNA samples were mixed and visualized via 1% agarose gel electrophoresis. The PCR products were purified with a kit and submitted to Guangzhou Gene Denovo Biotechnology for Illumina pyrosequencing.

### 2.5. Sequencing Data and Analyses of Diversity

The bacterial 16S rDNA and fungal ITS nucleotide sequences were assembled and filtered. Reads with ambiguous nucleotides, a quality score < 15, and those lacking a complete barcode and primer were deleted and excluded from further analysis, and the primer region was then removed. Usearch software (v.8.0.1623) [29] was used to cluster sequences and obtain operational taxonomic units (OTUs) at a 97% similarity level, and OTU anno-

tation was carried out based on the Silva (bacteria) (https://www.arb-silva.de accessed on 1 January 2020) and UNITE (fungi) (https://unite.ut.ee accessed on 1 January 2020) taxonomy databases. After removing non-bacterial or non-fungal OTUs, the OTU abundance information was normalized using the sequence number standard, which corresponded to the sample with the minimum sequence. Alpha diversity indexes, including the Shannon (https://mothur.org/wiki/shannon/ accessed on 1 January 2020), Simpson (https://mothur.org/wiki/simpson/ accessed on 1 January 2020), Chao (https://mothur.org/wiki/chao/ accessed on 1 January 2020), and ACE (https://mothur.org/wiki/ace/ accessed on 1 January 2020) indexes, were calculated using Mothur software (v.1.30).

For diversity analysis, the dissimilarity of soil bacterial and fungal communities under different crop patterns was calculated via principal coordinate analysis (PCoA), performed via pairwise Bray-Curtis dissimilarity using R software (version 2.14.0).

### 2.6. Statistical Analysis

The differences in soil attributes, soil microbial communities, and diversities under different crop patterns were tested using one-way analysis of variance (one-way ANOVA) via SPSS 22.0 (SPSS Inc., Chicago, IL, United States). Least significant difference tests were used to determine whether the differences between means were statistically significant. In all tests, $p < 0.05$ was considered statistically significant. The statistically significant differences of soil bacterial and fungal communities under the different crop patterns were performed through permutational multivariate one-way analysis of variance (PERMANOVA) and analysis of similarity (ANOSIM) based on Bray-Curtis dissimilarities. Spearman's correlation using R software (v.4.0.3) packages (stats v.3.3.5) was performed to analyze correlations between the diversity/richness of the soil communities and soil properties. Correlations between soil attributes and microbial community composition were assessed via redundancy analysis (RDA) using CANOCO for Windows 4.5.

### 3. Results

### 3.1. Soil Chemical Properties

Different crop pattern treatments had significant ($p < 0.05$) effects on soil OM ($F = 350.520$, $p = 0.000$), TN ($F = 93.084$, $p = 0.000$), TK ($F = 23.551$, $p = 0.000$), AP ($F = 9016.057$, $p = 0.000$), AK ($F = 3595.000$, $p = 0.000$), NN ($F = 8706.859$, $p = 0.000$), and pH ($F = 25.000$, $p = 0.000$) (Table 1). The soil OM, TN, NN, and pH under treatment T3 were significantly ($p < 0.05$) higher than those under the other three crop pattern treatments ($p = 0.000$) (Table 1). The soil TK and AP under treatment T2 were significantly ($p < 0.05$) higher than those under the other three crop pattern treatments ($p = 0.000$) (Table 1). However, the soil AK under treatment T1 was significantly ($p < 0.05$) higher than that under treatments T2, T3, and T4 ($p = 0.000$) (Table 1).

**Table 1.** The chemical properties of soil under different crop rotation treatments ($n = 4$, T1: grow watermelons for four years and then leave them fallow for three years, T2: grow watermelons for seven years, T3: grow watermelons for six years and wheat for one year, T4: crop pattern of one-year watermelon and one-year sunflower for seven years; T represented treatment of different crop pattern involved in this manuscript). Soil factors indicated include OM (organic matter), TN (total nitrogen), TP (total phosphorus), TK (total potassium), AP (available phosphorus), AK (available potassium), NN (nitrate-nitrogen), and pH.

| Treatment | OM (g/kg) | TN (%) | TP (%) | TK (%) | AP (mg/kg) | AK (mg/kg) | NN (mg/kg) | pH |
|---|---|---|---|---|---|---|---|---|
| T1 | 7.910 ± 0.069 b | 0.030 ± 0.001 b | 0.120 ± 0.000 | 1.910 ± 0.028 bc | 1.730 ± 0.023 d | 143.000 ± 0.408 a | 27.000 ± 0.141 d | 8.660 ± 0.011 b |
| T2 | 6.210 ± 0.026 d | 0.017 ± 0.000 c | 0.120 ± 0.004 | 2.090 ± 0.019 a | 10.500 ± 0.043 a | 120.000 ± 0.000 c | 36.000 ± 0.071 c | 8.550 ± 0.022 c |
| T3 | 8.420 ± 0.042 a | 0.053 ± 0.003 a | 0.130 ± 0.000 | 1.930 ± 0.023 b | 3.420 ± 0.036 c | 141.000 ± 0.408 b | 97.000 ± 0.319 a | 8.720 ± 0.015 a |
| T4 | 6.530 ± 0.076 c | 0.030 ± 0.001 b | 0.120 ± 0.004 | 1.850 ± 0.011 c | 4.460 ± 0.053 b | 107.000 ± 0.000 d | 41.000 ± 0.579 b | 8.620 ± 0.004 b |
| | *F*   *p* | *F*   *p* | *F*   *p* | *F*   *p* | *F*   *p* | *F*   *p* | *F*   *p* | *F*   *p* |
| T | 350.520   0.000 | 93.084   0.000 | 3.000   0.073 | 23.551   0.000 | 9016.057   0.000 | 3595.000   0.000 | 8706.859   0.000 | 25.000   0.000 |

Values are mean ± standard error ($n = 4$). The a–d mean significant difference $p < 0.05$ among different crop management treatments at 0.05 level.

### 3.2. Soil Bacterial and Fungal Communities

In this study, 14,905 bacterial and 2150 fungal OTUs were identified in all soil samples at a 97% sequence similarity cutoff (Figure S2). Bacterial communities of T1, T2, T3, and T4 shared 11,079 OTUs (Figure S2A), and fungal communities of T1, T2, T3, and T4 shared 420 OTUs (Figure S2B).

Soil bacterial communities were dominated by Proteobacteria, Planctomycetota, Actinobacteria, and Acidobacteria at the phylum level (Table S1). Different crop pattern treatments had significant ($p < 0.05$) influence on Proteobacteria ($F = 9.241$, $p = 0.000$), Gemmatimonadetes ($F = 6.618$, $p = 0.001$), Actinobacteria ($F = 21.750$, $p = 0.000$) and Acidobacteria ($F = 5.267$, $p = 0.005$) (Figure 1, Table S1). The relative abundance of Gemmatimonadetes and Actinobacteria was significantly ($p < 0.05$) higher in T1 than in other treatments; however, the relative abundance of Proteobacteria and Acidobacteria was significantly lower ($p < 0.05$) in T1 than in other treatments (Figure 1, Table S1).

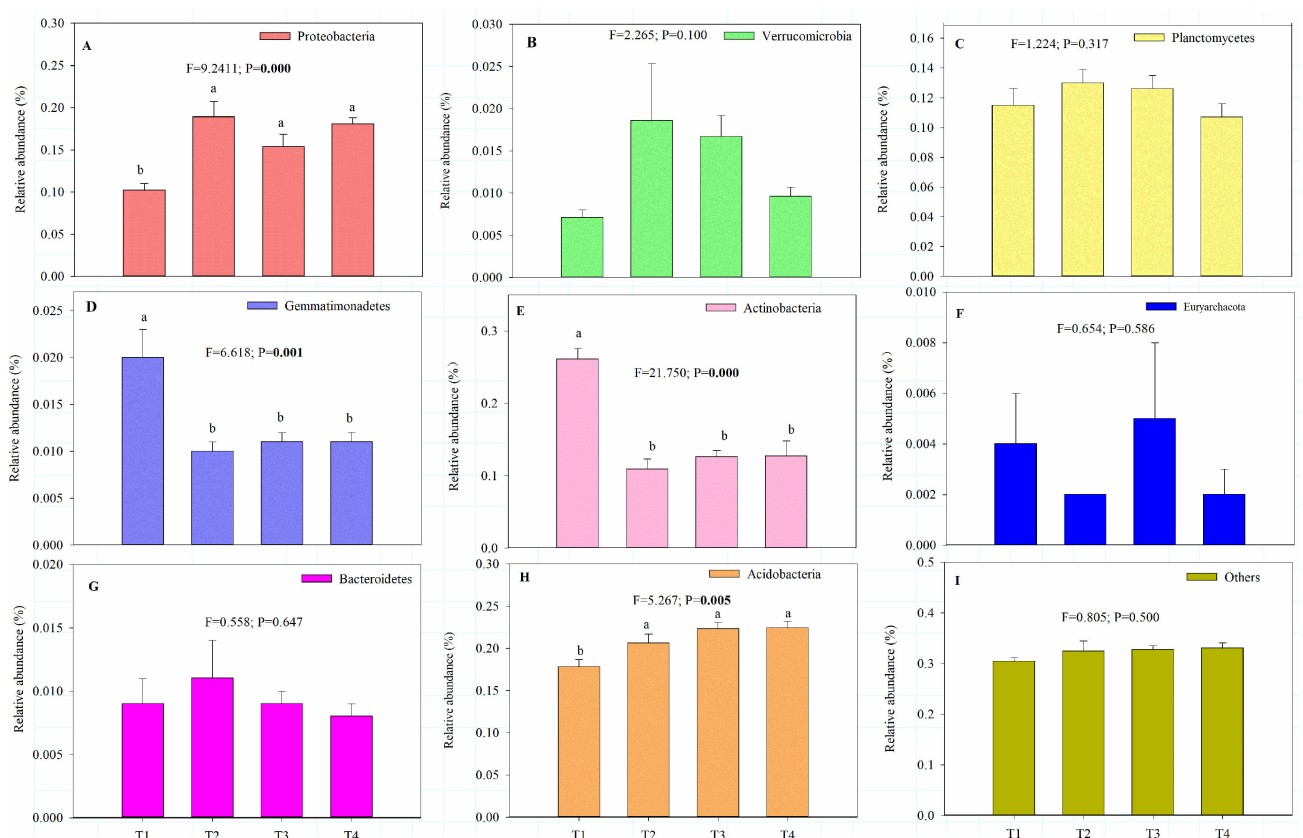

**Figure 1.** Taxonomic composition of soil bacterial community at the phylum level. Relative abundance of bacteria of all samples at the phylum level. ((**A**): Proteobacteria, (**B**): Verrucomicrobia, (**C**): Planctomycetes, (**D**): Gemmatimonadetes, (**E**): Actinobacteria, (**F**): Euryarchaeota, (**G**): Bacteroidetes, (**H**): Acidobacteria, (**I**): others. T1: grow watermelons for four years and then leave them fallow for three years; T2: grow watermelons for seven years; T3: grow watermelons for six years and wheat for one year; T4: grow watermelons for one year and rotate sunflower crop for one year. Values are mean ± standard error (SE), with bars indicating SE. Different lowercase letters indicate significant differences among the four treatments ($p < 0.05$), $n = 9$.

In addition, Ascomycota, Chytridiomycota, and Basidiomycota were the dominant phyla in the soil fungal community (Table S1), and different crop pattern treatments had significant ($p < 0.05$) influence on Ascomycota ($F = 3.259$, $p = 0.034$) and others ($F = 4.011$, $p = 0.016$) (Figure 2, Table S1). The relative abundance of Ascomycota was significantly ($p < 0.05$) lower in T1 than in T2 and T3, whereas that of others was significantly ($p < 0.05$) lower in T1 than in T2 and T3 (Figure 2, Table S1).

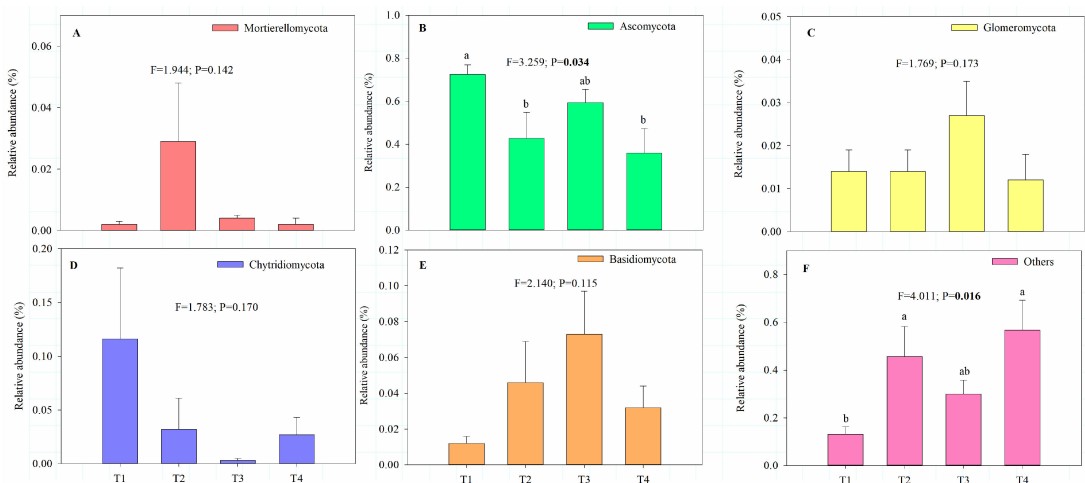

**Figure 2.** Taxonomic composition of soil fungal community at the phylum level. Relative abundance of fungi of all samples at the phylum level. ((**A**): Mortierellomycota, (**B**): Ascomycota, (**C**): Glomeromycota, (**D**): Chytridiomycota, (**E**): Basidiomycota, (**F**): others. Values are mean ± standard error (SE), with bars indicating SE. Different lowercase letters indicate significant differences among the four treatments ($p < 0.05$), $n = 9$.

### 3.3. Soil Bacterial and Fungal Community Diversities

For bacterial communities, the Chao and ACE richness indexes and the Shannon and Simpson diversity indexes under the four different crop patterns were not significantly different ($p > 0.05$) (Figure 3A,C,E,G).

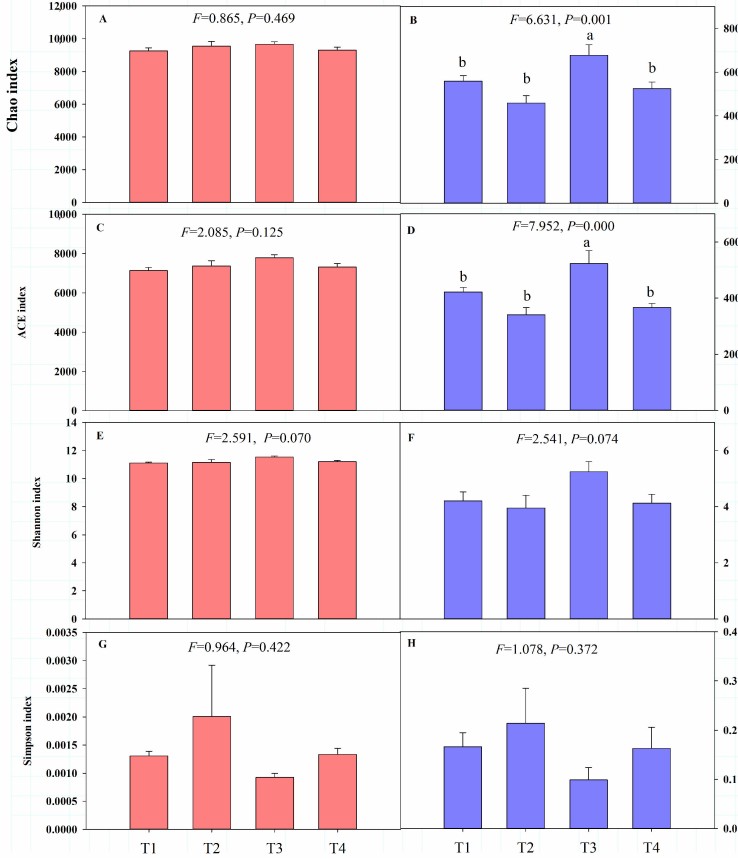

**Figure 3.** Soil microbial community alpha diversity index. Soil bacterial (**A,C,E,G**) and fungal (**B,D,F,H**) alpha diversity index. Different lowercase letters indicate significant differences among the four treatments ($p < 0.05$), n = 9.

In terms of fungal communities, the Chao and ACE richness indexes were significantly ($p < 0.05$) altered under different crop patterns (Figure 3B,D); however, the Shannon and Simpson diversity indexes were not significantly different ($p > 0.05$) among T1, T2, T3, and T4 ($p < 0.05$) (Figure 3F,H).

PCoA revealed that the soil microbial community diversity (including bacteria and fungi) was significantly different ($p < 0.05$) among T1, T2, T3, and T4 (Figure 4 and Table 2).

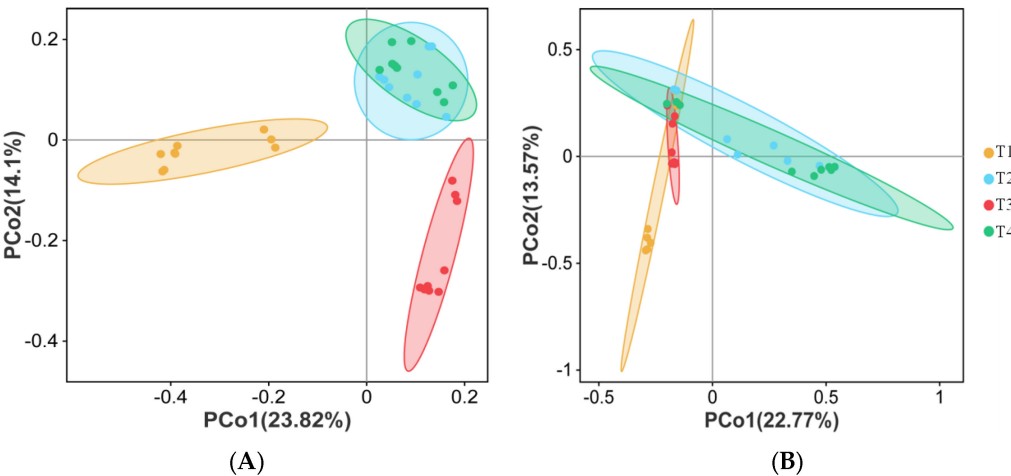

(A)          (B)

**Figure 4.** Principal coordinates analysis (PCoA) ordination. Bacteria (**A**) and fungi (**B**) of soil PCoA ordination based on Bray−Curtis dissimilarities at operational taxonomic units (OTU) level. (*n* = 9).

**Table 2.** The statistical test of analysis of similarity (ANOSIM) and permutational multivariate one-way analysis of variance (PERMANOVA) to analyze differences in soil bacterial and fungal community compositions measured by amplicon sequencing under different crop rotation treatments (*n* = 9).

| Type | df | PERMANOVA | | ANOSIM | |
|---|---|---|---|---|---|
| | | *F* | *p* | *R* | *p* |
| Bacteria | 3 | 5.017 | 0.0001 | 0.5702 | 0.0001 |
| Fungi | 3 | 3.27 | 0.0001 | 0.2584 | 0.0001 |

*3.4. Relationship between Soil Microbes and Soil Properties*

Spearman correlation results indicated that AP was significantly ($p < 0.05$) and positively associated with Simpson's index, while significantly ($p < 0.05$) and negatively associated with the diversity of the bacterial community of soil and Shannon index (Figure S3A). According to the RDA between the soil bacterial community and soil properties, the first and second axes of the RDA explained 56.75% and 4.20% of the variance, respectively, as the length of each arrow represents the contribution of the parameters to structural variation (Figure 5A). Notably, Proteobacteria were positively associated with soil TN, AP, and NN and negatively associated with soil TP, pH, AK, and OM (Figure 5A). Planctomycetota were positively associated with soil TP, TK, NN, AP, and TN and negatively associated with soil AK, OM, and pH (Figure 5A). Actinobacteria were positively associated with soil pH, OM, and AK and negatively associated with soil TN, AP, NN, TK, and TP (Figure 5A). Acidobacteria were positively associated with soil TN, AP, NN, TK, and TP and negatively associated with pH, OM, and AK (Figure 5A).

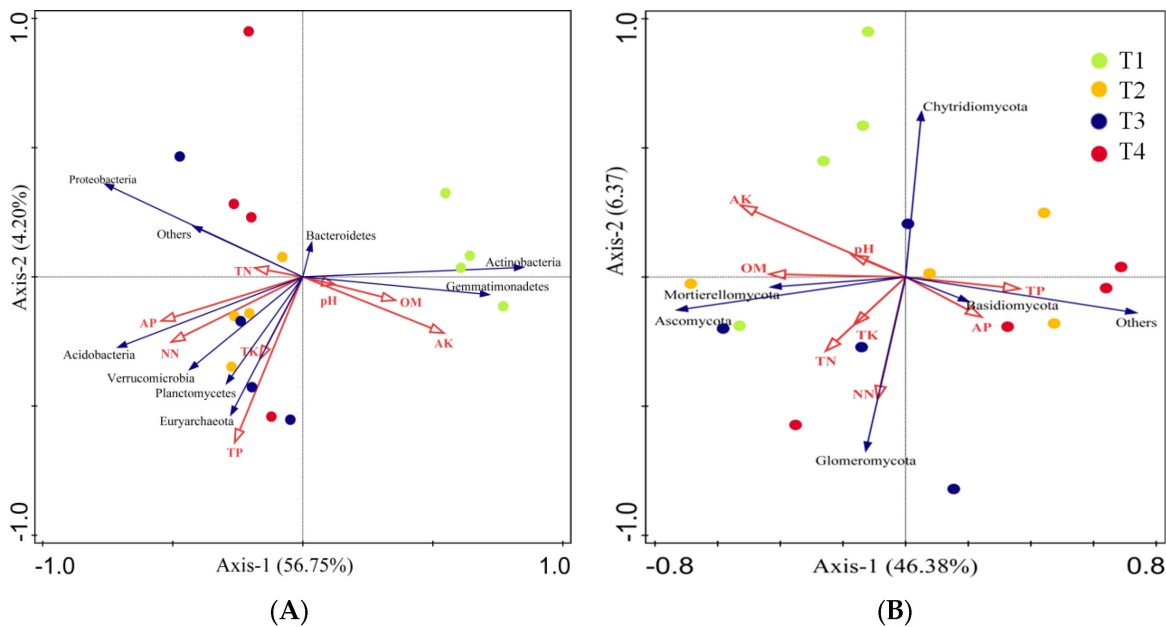

**Figure 5.** Redundancy analysis of relative abundance of soil bacterial (**A**) and fungal (**B**) communities and soil properties under different crop rotation treatments. Soil factors indicated include OM (organic matter), TN (total nitrogen), TP (total phosphorus), TK (total potassium), AP (available phosphorus), AK (available potassium), and NN (nitrate−nitrogen).

Spearman correlation results indicated that the richness and Shannon indexes of the fungal community of soil were significantly ($p < 0.05$) and positively associated with soil TP, TN, pH, NN, and OM, while Simpson's index of the fungal community of soil was significantly ($p < 0.05$) and negatively associated with soil TP, TN, pH, NN and OM (Figure S3B). According to the RDA between the fungal community and soil properties, the first and second axes of the plotted RDA results explained 46.38% and 6.37% of the variance, respectively (Figure 5B). Ascomycota was positively correlated with soil AK, pH, OM, TN, TK, and NN and negatively correlated with soil TP and AP (Figure 5B). Chytridiomycota were positively correlated with AK and pH but negatively correlated with AP, NN, TN, TK, and OM (Figure 5B). Basidiomycota was positively associated with TP, AP, and NN but negatively associated with soil TN, TK, pH, AK, and OM (Figure 5B).

## 4. Discussion

The present study investigated the effects of different crop planting patterns on soil chemical properties and microbial communities. A correlation analysis was conducted between soil microbes and properties, revealing many complex and close correlations between them.

### 4.1. Effects of Different Crop Planting Patterns on Soil Chemical Properties

Soil chemical properties play an important role in determining soil health [30]. Different typical and beneficial crop planting patterns have a positive impact on soil properties [31], such as increasing OM content [32,33]. Gikonyo et al. (2022) reported that the winter wheat-summer maize planting rotation had higher soil OM than other crop planting patterns [34]. Furthermore, the melon–cowpea intercropped system showed significantly increased TN, AP, and OM levels [35]. Lyu et al. (2020) revealed that OM, TK, and AP in cabbage and bean rotation treatments were significantly higher than those in the control continuous tomato cropping and that the pH and TN in celery rotation were significantly higher than those in continuous tomato cropping [36]. The results of the present study showed that the OM, TN, TK, AP, AK, NN, and pH concentrations were significantly affected by crop planting patterns. The six-year watermelon and one-year wheat rotation (T3) had the highest OM, TN, NN, and pH

concentrations than the other three treatments. This result supported the first hypothesis that the different crop planting patterns would affect the physical and chemical properties of the soil.

### 4.2. Effects of Different Crop Planting Patterns on Soil Microbial Community

Agricultural management practices, such as crop planting patterns, highly influence soil microbial ecosystems [37,38]. Understanding the effects of crop management treatments on agricultural soils and their microbial communities would have profound implications for improving agricultural production sustainability [39]. There is evidence of the effect of crop planting patterns on microbial diversity. Four rice rotation regimes (rapeseed, wheat, vegetables, and fallow) in agricultural ecosystems revealed bacterial families primarily classified into six phyla: Proteobacteria, Actinobacteria, Chloroflexi, Armatimonadetes, Nitrospirae, and Firmicutes [40]. Maize–wheat rotation revealed Proteobacteria as the most abundant phylum [41]. Bacterial communities were dominated by the Proteobacteria, Firmicutes, and Bacteroidetes phyla in melon crops grown in a closed hydroponic system [42]. Proteobacteria and Bacteroidetes dominated the soil bacterial communities of the melon–cowpea intercropping system [35]. Similarly, the results of the present study showed that the soil bacterial communities of the four crop management treatments were primarily dominated by Proteobacteria, Planctomycetota, Actinobacteria, and Acidobacteria at the phylum level (Figure 1) and that different treatments had a significant impact on these soil bacterial communities except Planctomycetota (Table S1).

Fungal communities play a central role in regulating soil fertility-related ecosystem services [43] because of their ecological functions, including carbohydrate-active enzyme production and OM breakdown, which allow the recycling and mobilization of mineral nutrients [44]. Liu et al. (2020) revealed that fungal communities were dominated by Ascomycota, Basidiomycota, and Zygomycota in long-term continuous soybean, maize, and wheat rotation systems [45]. Phyla were primarily dominated by Ascomycota and Basidiomycotain in melon crops grown in a closed hydroponic system [42]. Lyu et al. (2020) found that soil fungal communities were dominated by Ascomycota and Chytridiomycota after 6 years of continuous tomato cropping and rotation with cabbage, bean, or celery in greenhouse pots [36]. The results of this study are consistent with the findings of our study, which demonstrated that the soil fungal communities of the four crop treatments were primarily dominated by Ascomycota, Chytridiomycota, and Basidiomycota at the phylum level and that different treatments had a significant influence on fungal communities (Figure 2, Table S1). Along with previous findings, our second hypothesis that different crop planting patterns can change soil microbial communities is supported by the present study.

### 4.3. Effects of Different Crop Planting Patterns on Soil Microbial Diversity

Studies have provided evidence that microbial diversity can support ecosystem multifunctionality in various ways [46,47]. Soil microbial diversity has a positive effect on multifunctionality in agricultural soils [48,49]. Owing to the importance of microbial diversity in maintaining ecosystem functionality, researchers have focused on the relationships between crop management techniques and soil microbial diversity in agricultural ecosystems [47,50]. In the soil bacterial communities in the present study, the richness and diversity indexes of the seven-year watermelon crop planting pattern (T2) were the highest among the four rotations. Notably, these bacterial richness and diversity indexes were not significantly different between continuous watermelon cropping and other crop planting patterns. Studies have revealed that continuous cropping creates obstacles for farmers. This ultimately leads to yield reduction and quality decline when planting in a continuous cropping system, owing to pathogenic bacteria production [6,51]. In a seven-year experiment planting watermelons in a continuous cropping system in Zhongwei, China, we found that the bacterial diversity did not decrease (Figure 3). Thus, we can infer that long-term watermelon cultivation in Zhongwei City may be a suitable management strategy that does not adversely affect soil bacterial diversity.

Studies have demonstrated that soil fungal diversity increases with the number of years of peanut and strawberry cropping [10,52]. Bainard et al. (2017) revealed that two or more pulses, such as field peas (*Pisum sativum* L.), lentils (*Lens culinaris* Medik.), and chickpeas (*Cicer arietinum* L.), in 4-year crop planting patterns caused a significant decrease in fungal diversity as compared with continuous wheat planting or rotations with only one pulse crop [53]. Woo et al. (2022) found that planting peas alone caused a high reduction in fungal richness and diversity compared with wheat, pea–wheat rotation, and fallow [54]. In our study, we found that rotating six years of watermelon with one year of wheat (Treatment 3) exhibited the highest richness index than the other three rotations, and no significant difference was observed in diversity indexes among the other three rotations. Thus, the second hypothesis that different crop management treatments can change soil microbial communities is further supported.

### 4.4. Relationships between Microbial Community and Soil Properties

Soil properties can change the function and structure of microbial communities by allowing more suitable and adapted species to grow [30]. Studies addressing patterns of soil microbial communities have typically focused on arranging soil properties, such as pH [55] and OM/carbon content [56]. In this study, we analyzed the complex and close relationships between microbial communities and soil properties and found that Proteobacteria were positively associated with soil TN, AP, and NN and negatively associated with soil TP, pH, AK, and OM. Similarly, Actinobacteria and Acidobacteria were positively or negatively correlated with soil properties (Figure 5A). Our results are similar to those of previous studies. Dimitriu and Grayston (2010) demonstrated that relative Acidobacteria abundance increased with lower pH [57]. Shen et al. (2013) also found that bacterial community composition was closely related to soil pH [58]. Meyer et al. (2013) [59] demonstrated that AN regulates the relative diversity of bacteria in soil microbial communities in grassland ecosystems, and Nie et al. (2018) [60] revealed that the addition of high levels of nitrogen decreased soil bacterial diversity and altered the composition of the forest soil bacterial community.

A previous study revealed that fungal spore density is significantly related to soil pH and OM [61]. Indeed, previous studies have found that soil pH is a key factor in building microbial communities [55]. The most abundant taxon, Ascomycota, was strongly negatively correlated with soil pH and TN [36]. Although the Ascomycota fungal communities were negatively related to soil pH in this study, this suppression could be explained by a study that revealed that high pH reduced soil fungal diversity [62]. The results of our study also support the hypothesis that Ascomycota fungal communities were positively correlated with soil OM, AK, TN, TK, and NN and negatively correlated with soil TP and AP. These results revealed the close and complex relationships between soil microbial communities and soil physical and chemical properties.

This study investigated soil physical and chemical properties and microbial communities and confirmed the close and complex relationships between soil microbes and soil, enhancing our understanding and knowledge of watermelon crop patterns in Zhongwei, China.

## 5. Conclusions

This study revealed that different watermelon crop pattern treatments have significant effects on soil chemical properties. In addition, different crop patterns had a strong influence on the composition and diversity of soil bacterial and fungal communities. We found that their diversity and composition were closely related to the chemical properties of the soil. Compared with planting watermelons for seven years, the treatment of planting watermelons for six years and then planting wheat had a higher richness of fungal and bacterial communities. In future watermelon planting seasons, farmers can select the rotation pattern of melon and wheat to promote long-term preservation and health of the land. In addition, further research is needed to explore watermelon crop management techniques to improve the quality and yield of this crop in Zhongwei City, China.

**Supplementary Materials:** The following are available online at https://www.mdpi.com/article/10.3390/agriculture13051010/s1, Figure S1: Four different management fields; Figure S2: Venn diagram of soil bacterial (A) and fungal (B) OTUs; Figure S3: Spearman correlations of alpha diversity and soil properties; Table S1: Soil bacterial and fungal community composition at the phylum level.

**Author Contributions:** Conceptualization, M.T. and J.L.; methodology, S.L.; software, R.Y.; validation, M.T. and J.L.; formal analysis, S.L.; investigation, R.Y.; resources, X.Z.; data curation, X.Z.; writing—original draft preparation, J.L.; writing—review and editing, M.T. and J.L.; visualization, X.Z. supervision, X.Z.; project administration, M.T. and R.Y; funding acquisition, S.L. All authors have read and agreed to the published version of the manuscript.

**Funding:** This work was financially supported by the Leading Fund Project of Science and Technology Innovation of Ningxia Academy of Agricultural and Forestry Sciences (NKYG-22-03), Chinese Academy of Sciences 'Western Light' Talent Training Program ('Western Young Scholars') Project (XAB2022YW16), Ningxia Agricultural Science and Technology Independent Innovation Funding Project (NGSB-2021-7) and National Watermelon Industry Technology System Project (CARS-25).

**Institutional Review Board Statement:** Not applicable.

**Data Availability Statement:** Data is contained within the article.

**Acknowledgments:** We wish to thank the editor and anonymous reviewers for their valuable comments. Thanks to Michael Christensen from AgResearch Ltd., Grasslands Research Centre, New Zealand, for his valuable suggestions, meanwhile, we wish to thank Cory Matthew from School of Agriculture and Environment, College of Sciences, Massey University, Private Bag 11-222, Palmerston North, New Zealan, for his valuable comments.

**Conflicts of Interest:** The authors declare no conflict of interest.

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
