# Peer review of "Effects of Watermelon Cropping Management on Soil Bacteria and Fungi Biodiversity"

_agriculture, doi:10.3390/agriculture13051010_

Round 1

Reviewer 1 Report

With this study, Tian et al. are trying to investigate the impact of different crop rotations on soil microbial communities and soil properties in sandy fields used for cultivating selenium-rich sand watermelons. The researchers identified various bacterial and fungal phyla present in the soil and found that different crop rotations significantly affected the richness indices of fungal communities and several soil properties. The most effective rotation was found to be six years of watermelon followed by one year of wheat, which led to higher levels of organic matter, total nitrogen, nitrate nitrogen, and pH compared to other rotations. The continuous watermelon planting treatment, however, resulted in significantly higher levels of total potassium and available phosphorus. These findings suggest that incorporating crop rotations can play a crucial role in regulating soil microbial diversity and improving soil properties, ultimately enhancing the quality of watermelons grown in sandy soil.

The study is thoughtfully designed, and the methodology is clearly explained, making it easy for readers to follow the experiment and understand the results. However, I would recommend that the authors pay closer attention to the use of the English language throughout the manuscript. While the overall content is strong, there are instances where grammatical errors and awkward phrasing detract from the clarity of the text. By addressing these issues and carefully proofreading the manuscript, the authors can further enhance the readability and impact of their work.

Here are some minor concerns:

Line 113: Revise the sentence to read, "According to the different watermelon soil rotations, this study included four treatments."

Line 138: Change "-80°C" to "-80°C freezer."

Line 139: Change "4°C" to "4°C refrigerator."

Line 140: Apply the same change as in comment 5.

Line 202: Replace "significant effects" with "significant effects (P<0.05)."

Line 236: Change "between" to "among."

Line 239: Revise the phrase to read, "not significantly different (P<0.05)."

Line 368: Replace "N" with "nitrogen."

English grammar could be improved.

Reviewer 2 Report

Effects of watermelon cropping management on soil bacteria and fungi biodiversity

Manuscript Number: Agriculture-2365371

In my opinion, the subject matter dealt with by the authors is very interesting, because of improving soil fertility and microbial activity through crop rotation instead of monocropping is a great idea. However, having thoroughly reviewed the manuscript presented to me, I have some major comments and suggestions, which I present below:

1.     The manuscript suffers from lots of grammatical and spelling mistakes.

2.     Why the field was kept fallow for three years in treatment T1?

3.     What is difference between treatment T2 and T3? They are same just difference in one year cultivation.

4.     In line 76, the author mentioned that there will be a crop rotation of watermelon and sunflower in one system and water melon and wheat in another cropping system. Cultivation of watermelon for six consecutive years and wheat in one year is not a changing crop rotation. This is bit confusing.

5.     The initial soil data is missing which is very crucial for this study.

6.     What was the land use history of this site before starting the experiment?

7.     There should be an initial soil analysis data before starting the experiment including microbial community structure to understand the effect of cropping pattern and land management over the seven years. Otherwise, it is very difficult to understand real effect of land management on the soil OM and microbial activity.

8.     Why there is no lettering in Fig. 1 B?

9.     Table 2 is difficulty to understand. Why only T?

10.  In objectives 1 and Conclusion, the author mentioned about soil physical properties which is completely missing in this study.

11.  In Conclusion, the author mentioned that different crop rotation system but they used only one crop rotation system.

12.  The Conclusion is very poorly written and out of focus. It is obvious that various cropping pattern will show variability in microbial activity due to their root system. In many cases the monocropping showed better performance that multicopying? What is the key message of this study?   

The manuscript suffers from lots of grammatical and spelling mistakes and needs careful editing

Reviewer 3 Report

The article provides a good background on the topic of watermelon cultivation and its impact on soil fertility, which helps to contextualize the study. The research question is clearly stated, and the methodology section describes the techniques used to explore the effects of different crop rotations on soil microbial communities and soil properties. The results section provides a comprehensive summary of the findings, including the different bacterial and fungal phyla found in the soil, the effect of crop rotations on soil properties, and the relationships between soil properties and soil microbial communities. Overall, this research article is well-written and effectively communicates the findings of the study and acceptable in current form.
